# F-3DGS: Factorized Coordinates and Representations for 3D Gaussian Splatting

## ABSTRACT

The neural radiance field (NeRF) has made significant strides in representing 3D scenes and synthesizing novel views. Despite its advancements, the high computational costs of NeRF have posed challenges for its deployment in resource-constrained environments and real-time applications. As an alternative to NeRF-like neural rendering methods, 3D Gaussian Splatting (3DGS) offers rapid rendering speeds while maintaining excellent image quality. However, as it represents objects and scenes using a myriad of Gaussians, it requires substantial storage to achieve high-quality representation. To mitigate the storage overhead, we propose Factorized 3D Gaussian Splatting (F-3DGS), a novel approach that drastically reduces storage requirements while preserving image quality. Inspired by classical matrix and tensor factorization techniques, our method represents and approximates dense clusters of Gaussians with significantly fewer Gaussians through efficient factorization. We aim to efficiently represent dense 3D Gaussians by approximating them with a limited amount of information for each axis and their combinations. This method allows us to encode a substantially large number of Gaussians along with their essential attributes—such as color, scale, and rotation—necessary for rendering using a relatively small number of elements. Extensive experimental results demonstrate that F-3DGS achieves a significant reduction in storage costs while maintaining comparable quality in rendered images.

## CCS CONCEPTS

• **3D novel view synthesis → 3D Gaussian Splatting**; • **Computer Vision**;

## KEYWORDS

3D Gaussian Splatting, 3D Reconstruction, Real-Time Rendering, Tensor Factorization, Compression

## 1 INTRODUCTION

The Neural Radiance Field (NeRF) [25] has demonstrated substantial success in representing 3D scenes through its differentiable volumetric rendering technique and high representation quality. This method has found widespread application in scenarios ranging from novel view synthesis [1, 2, 25, 38, 40], 3D object generation [30], editing [15, 16, 35], segmentation [6, 23], and navigation [36], to name a few. As this representation method adopts the concept of a

Permission to make digital or hard copies of all or part of this work for personal or classroom use is granted without fee provided that copies are not made or distributed for profit or commercial advantage and that copies bear this notice and the full citation on the first page. Copyrights for components of this work owned by others than the author(s) must be honored. Abstracting with credit is permitted. To copy otherwise, or republish, to post on servers or to redistribute to lists, requires prior specific permission and/or a fee. Request permissions from permissions@acm.org.
*ACM MM, 2024, Melbourne, Australia*
© 2024 Copyright held by the owner/author(s). Publication rights licensed to ACM.
ACM ISBN 978-x-xxxx-xxxx-x/YY/MM
https://doi.org/10.1145/nnnnnnn.nnnnnnn

'field' from physics, in which each point possesses a corresponding value, it requires sampling and processing values from numerous points for each pixel to compute the pixel's color. This can pose significant challenges, especially when attempting real-time rendering on systems with limited resources [10].

Rasterization-based methods, particularly 3D Gaussian Splatting (3DGS) [17], have emerged as alternatives within the differentiable rendering frameworks. Distinct from NeRF-like methods using hundreds of samples per pixel for volumetric rendering, 3DGS circumvents the need to sample values from empty spaces. This efficiency stems from its primitive-based rendering approach, enabling it to achieve extremely high rendering speed without sacrificing image quality. 3DGS represents a scene with 3D Gaussians as the geometric primitives, learning each Gaussian attribute, such as color, scale, rotation, and opacity, under a fully differentiable rendering pipeline. Through millions of Gaussians, the approach has shown that it can capture detailed textures and nuances, producing exceptional quality, while enjoying the lower computational complexity of rendering via geometric primitives.

Despite the fast rendering speed and quality enhancements brought by 3DGS, it comes with several significant drawbacks. A primary challenge is its dependence on an extensive number of 3D Gaussians to maintain high image fidelity. Rendering a high-quality image of a detailed real-world scene often involves several million Gaussians, adding to the system's spatial complexity. Furthermore, each Gaussian is characterized by multiple coefficients to ensure high-quality rendering results. Specifically, it entails 48 spherical harmonics coefficients for color, one for opacity, three for scale, and four quaternion coefficients for rotation. Due to the above issues, 3DGS requires large memory and storage footprints, and this limitation is especially evident in complex, unbounded scenes, such as those in Mip-NeRF360 datasets. In addition, the computational complexity of the current 3DGS rendering algorithm is proportional to the number of 3D Gaussians. Thus, a larger number of Gaussian results in reduced rendering speed.

The aforementioned inefficiency stems from the inherent inability of 3DGS to utilize structural patterns or redundancies. We observed that 3DGS produces an unnecessarily large number of Gaussians even for representing simple geometric structures, such as flat surfaces. Moreover, nearby Gaussians sometimes exhibit similar attributes, suggesting the potential for enhancing efficiency by removing the redundant representations.

This paper presents a novel method to overcome the limitations of 3DGS, employing structured coordinates and decomposed representations of Gaussians through factorization, as shown in Fig. 1. Our approach took inspiration from classical tensor or matrix factorization techniques, which have also been extensively investigated in the NeRF literature [7, 9, 11, 13]. We propose a factorized coordinate scheme in which we maintain 1D or 2D coordinates in each axis or plane and generate 3D coordinates by a tensor product.

This approach allows us to generate numerous 3D coordinates of Gaussians only with a small number of 1D or 2D coordinates, significantly improving spatial efficiency for the position parameters. Furthermore, the proposed factorization method extends beyond the spatial coordinates to include related attributes, such as color, scale, rotation, and opacity. By exploiting structural patterns and redundancies, the factorized Gaussian attributes can efficiently compress the model size while preserving each Gaussian's essential characteristics.

Based on the factorized coordinates and attributes, some of the expanded Gaussians through de-factorization could be non-essential for rendering quality (e.g., de-factorized Gaussians in empty spaces). Motivated by a recent work [21], we use a binary mask for removing unnecessary Gaussians to accelerate both training and rendering speed. During training, we update the mask using the straight-through-estimator technique. Once training is complete, we only need to keep the binary values. As the mask only requires 1 bit per value, it incurs negligible storage overhead. Using the binary mask not only marginally improves representation quality but also significantly increases rendering speed, nearly doubling the frames per second (FPS) on the NeRF synthetic dataset.

In this study, we conducted extensive experiments to validate the effectiveness of our approach in reducing the spatial redundancy inherent in 3DGS. These experiments demonstrate that our factorization method significantly reduces the required storage by downsizing 3DGS over 90% while achieving comparable image quality. We also provide qualitative results and analysis to support our findings and conclusions, helping to provide a comprehensive understanding of the subject matter. Finally, we prove our approach as an efficient framework for representing 3D scenes, achieving high performance, compact storage, and fast rendering.

## 2 RELATED WORK

### 2.1 Neural Rendering

Neural rendering[1] is an emerging approach that leverages machine learning to generate photorealistic visual objects and scenes. Central to this domain is differentiable rendering, enabling the end-to-end training and optimization of 3D scene representations directly from images [20]. Along with differentiable rendering, volumetric rendering has also been an important component in neural rendering. As addressed in 3DGS [17], the general volumetric rendering equation to calculate the color value $C$ of each pixel can be represented as follows:

$$C = \Sigma_{i=1}^{N} c_i \alpha_i \Pi_{j=1}^{i-1}(1 - \alpha_j), \quad (1)$$

where $c_i$ and $\alpha_i$ represent the color and opacity of each sampled point, respectively. The specifics of how points are sampled, the number of samples per pixel $N$, and the method for calculating opacity $\alpha_i$ can vary across different rendering methods.

A seminal work in this field is Neural Radiance Fields (NeRF) [25], which adopts deep neural networks and positional encoding to represent volumetric scene features for high-fidelity rendering [25]. NeRF interprets objects and scenes as fields where every point in

space possesses specific values that are computationally generated by neural networks. These values are then rendered using the volumetric rendering equation (Eq. 1). While NeRF allows for detailed rendering, it necessitates dense sampling of the entire volume, including empty spaces, which markedly increases computational costs. Several strategies have been proposed to mitigate this limitation [10, 39], yet achieving both high-quality representation and real-time rendering remains a significant challenge.

Addressing the inefficiencies in NeRF's approach, 3D Gaussian Splatting (3DGS) focuses on rendering only known primitives (Gaussians), thereby eliminating the need for sampling in empty areas and greatly reducing computational costs [17]. However, this technique requires a substantial number of Gaussians to maintain high-quality scene representations, leading to increased storage usage—sometimes more than 1GB for unbounded scenes [17]. Our research, F-3DGS, builds upon 3DGS, targeting its main drawbacks: the high memory requirement and the large number of parameters needed for high-quality rendering. F-3DGS proposes a parameter-efficient approach that supports fast training and real-time rendering capabilities, making high-quality neural rendering more feasible for widespread applications.

### 2.2 Factorization Techniques for Neural Rendering

To mitigate the heavy computational costs of NeRF, several works have proposed incorporating additional data structures while retaining the volumetric rendering process. Among these, Plenoxels [12] and Plenoctrees [39] proposed utilizing three-dimensional data structures. However, these approaches were highly inefficient in terms of storage requirements, as the number of required parameters increases cubically with the resolution per axis. To address this, various studies have explored more compact and efficient representations by decomposing 3D structures into combinations of lower-dimensional elements. Specifically, TensoRF [9] and Strivec [13] have implemented the canonical polyadic (CP) decomposition, which uses one-dimensional lines to approximate higher-dimensional tensors, thereby enhancing the compactness of 3D representation. TensoRF further extended this approach with vector-matrix (VM) decomposition, while other studies such as EG3D [7], Hex-Planes [5], and K-Planes [19] have focused on matrix-only decompositions. Nevertheless, these methods have not achieved real-time rendering due to the fundamental computational limitations of the NeRF-like rendering process. In contrast, our work integrates the concept of factorization with 3DGS, thus enjoying the dual benefits of fast rendering speeds and substantially reduced storage costs. Our method is closely related to Strivec, as it decomposes a scene into a group of small areas, each represented through efficient decomposition. The principal differences lie in factorizing irregular point representations rather than using regular grid representations, which allows us to leverage the fast rasterization-based rendering pipeline. Experimental results demonstrate that our proposed method achieves rendering speeds more than five times faster than Strivec while maintaining similar or superior representation quality.

---

[1]We use the "neural rendering" term to refer to a general rendering technique that exploits differentiable computational pipelines to render images, which includes both NeRF and 3DGS.

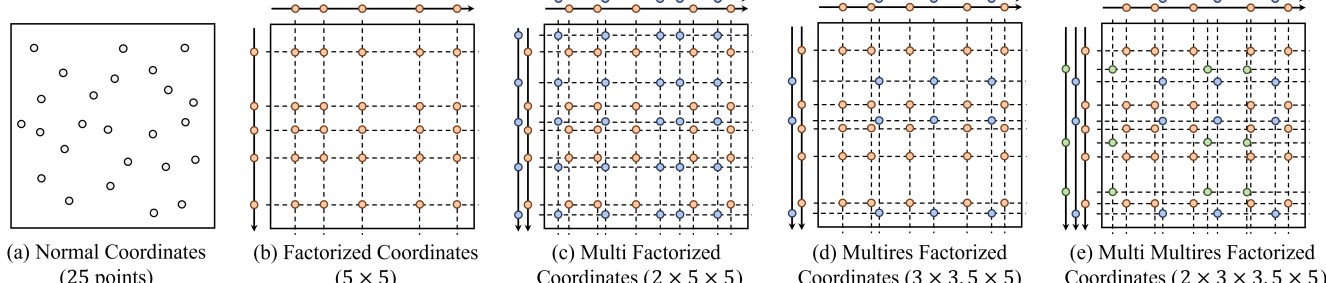

| (a) Normal Coordinates (25 points) | (b) Factorized Coordinates (5 × 5) | (c) Multi Factorized Coordinates (2 × 5 × 5) | (d) Multires Factorized Coordinates (3 × 3, 5 × 5) | (e) Multi Multires Factorized Coordinates (2 × 3 × 3, 5 × 5) |

**Figure 1: Examples of factorized coordinates: (a) 25 normal coordinates, (b) 5 × 5 factorized coordinates. each $x$ and $y$ axis has 5 points, and both represent 25 (5 × 5) points. (c) two 5 × 5 factorized coordinates and a total of 50 points are represented (2 × 5 × 5), (d) multi-resolution factorized coordinates, where two factorized coordinates have different resolutions (3 × 3 and 5 × 5), represent total 34 points, (e) two 3 × 3 and one 5 × 5 factorized coordinates. A total of 43 points are represented. The best-viewed in color.**

## 2.3 Lightweighting 3D Gaussian Splatting

Inspired by the real-time rendering and high quality of 3DGS, numerous studies have been conducted to maintain these advantages while lightweighting the method for broader applications. One of the key strategies for achieving a lighter-weight 3DGS involves minimizing the number of Gaussians used. Several studies demonstrated that techniques such as pruning [21, 28] and the use of anchors [24, 31] can significantly reduce the number of Gaussians without degrading image quality.

Beyond simply reducing the number of Gaussians, some studies aim to minimize the average size required to represent each Gaussian. This includes employing lower-bit precision, codebooks, quantization, and compression algorithms like entropy encoding, which collectively make the overall size compact [14, 21, 27].

Our approach integrates both of these strategies, aiming to reduce not only the number of Gaussians needed but also the data size required for representing each Gaussian. Our research is closely related to structured 3DGS approaches [24, 31]. We hypothesize that Gaussians located in proximity can be organized into structured representations and further compacted via factorization techniques. Our method significantly lowers the storage requirements for 3DGS, enhancing its applicability in scenarios demanding both high-quality rendering and storage efficiency.

## 3 FACTORIZED 3D GAUSSIAN SPLATTING

In this section, we present our method to make the 3D Gaussian splatting model much lighter through factorization of coordinates (Sec. 3.2) and features (Sec. 3.3). Our method integrates two distinct factorization methods: canonical polyadic (CP) and vector-matrix decompositions [9, 13]. Starting with the background, we delve into the specifics of each factorization strategy in the subsequent sections.

## 3.1 Background: 3D Gaussian Splatting

3D Gaussian splatting is a method that employs 3D Gaussians as primitives for representing 3D scenes. The differentiability of 3D Gaussians and their ease of projection onto 2D image planes facilitate efficient $\alpha$-blending in the rendering process [41]. Each Gaussian has several features, including color (represented by spherical harmonics (SH) coefficients), scale, rotation, and opacity parameters. The method uses a collection of these Gaussians to represent the whole scene. The entire rendering pipeline is differentiable, which allows end-to-end training of all Gaussian features given only the images from various views.

The initialization process begins with a Structure from Motion (SfM) algorithm [33] to generate sparse point clouds. These serve as initial estimates for the Gaussian positions. The densification stage follows, where the number of Gaussians is incrementally increased through cloning and splitting, enhancing scene details. The algorithm starts with a coarse global structure and systematically integrates finer details by adding more Gaussians. Upon reaching a set number of training iterations, a final fine-tuning phase optimizes all parameters while maintaining a constant Gaussian count. This approach has demonstrated state-of-the-art image quality and exceptional rendering speeds. For further details and comprehensive information, please refer to the original paper [17].

## 3.2 Factorized Coordinates

*3.2.1 CP Factorized Coordinates.* Drawing inspiration from the canonical polyadic (CP) decomposition method to neural rendering [9, 13], we introduce factorization for representing coordinates. Let $p_x = \{x_1, \ldots, x_N\}$, where $x_i \in \mathbb{R}$, be a set of factorized coordinates in $x$-axis. $N$ is the number of points on the axis ($|p_x| = N$). Similarly, we can define factorized coordinates for the $y$ and $z$ axes as $p_y = \{y_1, \ldots, y_N\}$ and $p_z = \{z_1, \ldots, z_N\}$, respectively.

For these factorized coordinates, we can express the entire set of points in 3D spaces as follows:

$$p_{xyz} = p_x \times p_y \times p_z = \{(x_1, y_1, z_1), \ldots, (x_N, y_N, z_N)\}, \quad (2)$$

where '×' denotes the cartesian product of two sets. Then, the total number of points in $p_{xyz}$ is $N^3$, and considering that each point requires three real numbers to represent its coordinate, it needs a total of $3N^3$ numbers. However, if those points are aligned, we can compactly represent $N^3$ points with only $3N$ numbers, using only $N$ points for each axis. That is, for well-aligned coordinates, factorized

coordinate representations can reduce the number of parameters by $1/N^2$. This can substantially alleviate the spatial complexity associated with storing position information. For example, 1,000 points per axis can cover up to 1 billion points using only 3,000 numbers. As illustrated in Fig. 1-(b), however, this approach is only capable of constructing a very restricted range of structural shapes.

Therefore, we propose a hybrid approach in which we utilize multiple sets of factorized coordinates with a small $N$ instead of having a single set of factorized coordinates with a large $N$ (Fig. 1-(c)). In this approach, we employ $B$ sets of factorized coordinates, where each set, denoted as $p_{xyz}^b = p_x^b \times p_y^b \times p_z^b$, can represent a total of $N^3$ points, and consequently constructing $BN^3$ number of points with $3BN$ number of points. While the compression ratio remains the same as $1/N^2$, this approach demands more parameters due to the smaller values for $N$ (usually $N \leq 10$ to obtain a good rendering quality). Furthermore, each set of factorized coordinates can have varying resolutions, as illustrated in Fig. 1-(d) and (e), thereby offering more flexibility in the positions, which increases the expressibility. Considering the multi-set and multi-resolution factorized coordinates, we can rewrite the set of whole points constructed by the proposed method as,

$$p_{xyz} = \bigcup_{b=1}^{B} p_x^b \times p_y^b \times p_z^b, \tag{3}$$

where $p_x^b = \{p_1^b, \ldots, p_{N_b}^b\}$, and $N_b$ denotes the number of point in each axis of $b$-th factorized coordinate set. That is, using only $3\sum_b N_b$ numbers, we can represent at maximum $\sum_b N_b^3$ three-dimensional coordinates. As shown in Fig. 1, we can approximate dense Gaussians with a small number of factorized coordinates.

*3.2.2 VM Factorized Coordinates.* The recently introduced plane-based decomposition methods, such as TensoRF [9] and K-planes [11], have achieved promising results, offering parameter-efficient representations in NeRF. In a similar vein, our approach incorporates a vector-matrix (VM) factorization [9] for coordinates. Let $p_{xy} = \{(x_1, y_1), \ldots, (x_N, y_N)\}$ (we omitted the superscript $b$ for brevity onwards) be a set of two-dimensional coordinates in the $xy$ plane, and $p_z = \{z_1, \ldots, z_N\}$ is a set of one-dimensional coordinates in the $z$ axis (similarly, we can also define $p_{yz}, p_x, p_{xz}$, and $p_y$). Then, the set of whole points in the VM factorized coordinate scheme is defined as

$$p_{xyz} = (p_{xy} \times p_z) \cup (p_{yz} \times p_x) \cup (p_{xy} \times p_y). \tag{4}$$

The points on the plane, e.g., $p_{xy}$, can move freely without any constraints, increasing the flexibility of positions at the expense of the higher spatial complexity required for storing two-dimensional coordinates.

## 3.3 Factorized Representations

*3.3.1 CP Factorized Representations.* The primary factor contributing to the storage requirements of 3DGS is the rich sets of parameters of each 3D Gaussian. In this section, we describe the factorized representation of those attributes for each Gaussian without compromising the rendered image quality. Similar to the CP factorized coordinate scheme, we hold the features associated with the factorized coordinates in each axis. For example, the $x$ axis scale

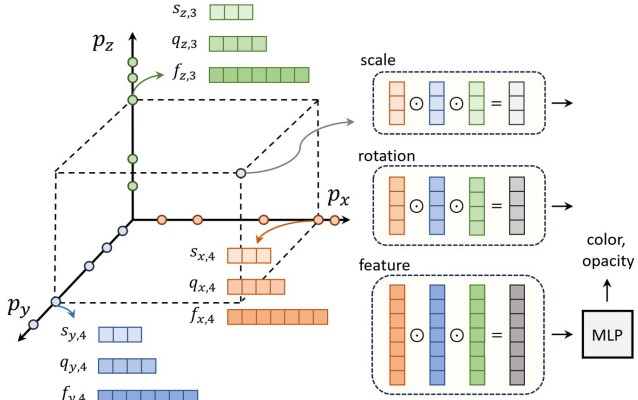

**Figure 2: Illustration of factorized coordinates and representations. $p$, $s$, $q$, and $f$ denote coordinate, scale, rotation (in quaternion), and features for color and opacities, respectively. The lower indices of $s$, $q$, and $f$ are the axis and the indices of the feature dimension. For element-wise multiplication, we used the $\odot$ notation.**

parameters for each Gaussian can be obtained by the element-wise product of the features in the factorized coordinates. More formally,

$$s_{xyz,i} = s_{x,i} \circ s_{y,i} \circ s_{z,i}, \tag{5}$$

where $\circ$ represents the outer product, $s_x, s_y, s_z \in \mathbb{R}^{N \times 3}$ denotes the scale parameters for $N$ points along the $x, y, z$ axis in the factorized coordinates respectively, and $s_{x,i} \in \mathbb{R}^N$ indicates the $i$-th parameters in the scale parameters. $s_{xyz} \in \mathbb{R}^{N \times N \times N \times 3}$ is the final scale parameters used in the rendering process, $s_{xyz,i} \in \mathbb{R}^{N \times N \times N}$ denotes the $i$-th parameters. Each element in $s_{xyz}$ corresponds to the scale parameter associated with the point in $p_{xyz}$. We also factorize rotation parameters $q_{xyz} \in \mathbb{R}^{N \times N \times N \times 4}$ in the same way.

To handle view-dependent color features and opacity, we employ a Multi-Layer Perceptron (MLP) to generate coefficients for spherical harmonics and opacity. It takes low-dimensional learnable features as input, and we factorize these learnable features along each axis. We maintain a factorized feature vector for colors and opacity for each axis and compute the input feature for the MLP as follows.

$$f_{xyz,i} = f_{x,i} \circ f_{y,i} \circ f_{z,i}, \tag{6}$$

$$c_{xyz}, \alpha_{xyz} \leftarrow \text{MLP}(f_{xyz}; \theta), \tag{7}$$

where $f_x, f_y, f_z \in \mathbb{R}^{N \times d}$ denote the learnable feature vectors for $N$ points along the $x, y, z$ axes respectively, and $d$ is the learnable feature dimension. Then, we obtain the features of the whole set of points $f_{xyz} \in \mathbb{R}^{N \times N \times N \times d}$, and the MLP takes the features $f_{xyz}$ as a batch and generates color features $c_{xyz} \in \mathbb{R}^{N \times N \times N \times 48}$ (three degrees of spherical harmonics for example), and opacity $\alpha_{xyz} \in [0, 1]^{N \times N \times N}$.

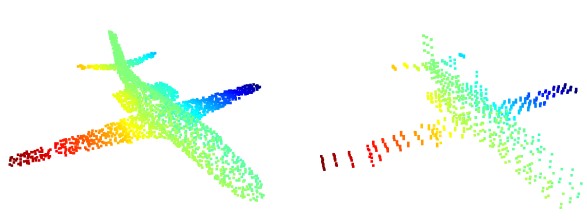

(a) Ground truth point clouds    (b) Factorized point clouds

Figure 3: Visualization of factorized coordinate sets. The right figure shows the approximation of 2,488 three-dimensional coordinates using only 30 factorized coordinate sets with an $N_b$ of three.

*3.3.2 VM Factorized Representations.* We also introduce VM feature decomposition for the VM factorized coordinates. The generated color features and opacity are written as follows.

$$f_{xyz,i} = f_{xy,i} \circ f_{z,i} + f_{yz,i} \circ f_{x,i} + f_{xz,i} \circ f_{y,i}, \quad (8)$$

$$c_{xyz}, \alpha_{xyz} \leftarrow \text{MLP}(f_{xyz}; \theta), \quad (9)$$

where $f_{xy}, f_{yz}, f_{xz} \in \mathbb{R}^{N \times N \times d}$ are the feature matrices associated with the VM factorized coordinates $p_{xy}, p_{yz}, p_{xz}$. In addition, $f_x, f_y, f_z \in \mathbb{R}^{N \times d}$ represent the feature vectors associated with $p_x, p_y, p_z$, respectively.

## 3.4 Initialization

The initialization scheme plays a pivotal role in achieving high rendering quality in 3D Gaussian Splatting (3DGS), especially for complex objects and scenes [17]. The original 3DGS leverages a structure from motion technique [33], to initialize 3D Gaussian points at the onset of training.

In our proposed method, we observed a similar initialization effect on the resulting models, particularly for the factorized coordinates. To address this, we propose a heuristic method to effectively determine the initial positions of 3D Gaussians. Our approach begins by generating point clouds from a pre-trained 3DGS model of a target scene.

First, we identify the boundary values of the point cloud as $x_{max}$, $x_{min}, y_{max}, y_{min}$ and $z_{max}, z_{min}$. After multiplying the bound value by 1.2, we divide it into multiple sets of bins by average interval. For example, we set 0.026 in the nerf-synthetic dataset, and the number of bins along the x-axis will be $1.2 * (x_{max} - x_{min})/0.026$. After calculating a 3D histogram from the point cloud coordinates for the set of bins, we construct a set of factorized coordinates (each set of factorized coordinates has $N^3$ points, $N$ points along each axis, we omitted the superscript '$b$') in the histogram bin which contains several points exceeding the threshold $\lambda$. For example, we set 5 for synthetic-NeRF dataset [25], and discard the bins that have no more than 5 points. Furthermore, for bins that have more than $N^3$ points, more factorized coordinates are required, so we introduce an additional set of factorized coordinates for every additional $N^3$ points.

This heuristic method has proven effective in practice. We will present a comparison of our initialization scheme with random initialization in Tab. 4, highlighting its practical efficacy.

## 3.5 Masking

As addressed in Sec. 3.2.1 and 3.3.2, factorized coordinates and representations can be highly efficient in terms of compactness. However, using all $N^3$ is computationally expensive since the current rasterization pipeline in 3DGS involves linear sorting and $\alpha$-blending to render an image; hence, the computational complexity is proportional to the number of Gaussians. Furthermore, we also observed many Gaussians in the factorized coordinates do not contribute to the final rendering results (e.g., when the opacity $\alpha < 0.001$). Therefore, pruning the less important 3D Gaussians can readily accelerate both training and rendering processes. Inspired by the recent works [21, 32] in compact representations for NeRFs and 3D Gaussian Splatting, we adopt binary masks to prune out non-used coordinates while exploiting the compactness of factorized representations. To train binary masks, we used the straight-through-estimator method [4] to update mask values. Having a trainable mask for each coordinate requires total $N^3$ values. However, once the binary mask is trained, a single bit per coordinate is sufficient. Therefore, the spatial overhead of using binary masks is negligible.

We applied masks to variables that are directly related to visibility: both colors $c_{xyz}$ and opacities $\alpha_{xyz}$. Since there are total $N^3$ points that can be represented using factorized coordinates (Sec. 3.2.1), we also need $N^3$ binary masks $M \in \mathbb{R}^{N \times N \times N}$ for each coordinate set (To avoid the notational clutter, we omitted the superscript '$b$', which denotes a set of factorized coordinates.). During training, we use floating points for masks, as we cannot properly calculate gradients for binary variables. We use the following equation for the binarized masks $\bar{M}$ during training:

$$\bar{M} = \text{sg}(\mathcal{H}(M - \tau) - \sigma(M)) + \sigma(M), \quad (10)$$

where sg, $\mathcal{H}$, $\tau$, and $\sigma$ denote the stop gradient function, Heaviside function, the threshold, and the element-wise sigmoid function, respectively. Then, we element-wisely multiplied this binarized mask $\bar{M}$ to scales $s_{xyz,i} \in \mathbb{R}^{N \times N \times N}$ and opacities $\alpha_{xyz} \in [0,1]^{N \times N \times N}$. To increase the sparsity of the mask for more efficient representations, we add a regularizing loss term to the objective function so that mask values lean towards being zeroed out. The loss is defined as follows:

$$\mathcal{L}_m = \sum_i \sum_j \sum_k \sigma(M_{ijk}), \quad (11)$$

where $M_{ijk} \in \mathbb{R}$ denotes an element of the mask.

## 4 EXPERIMENT

### 4.1 Proof-of-Concept Experiment

In this section, we introduce a proof-of-concept experiment designed to demonstrate the feasibility of our method using point clouds. The experiment involves initializing sets of factorized coordinates and optimizing them using the Chamfer distance function. The primary objective here is to showcase the effectiveness and efficiency of factorized coordinates in approximating ground truth point clouds. For this experiment, we selected a sample point cloud of an object from ShapeNetCorev2 [8].

Table 1: Comparison of our method with the previous and concurrent novel view synthesis methods on synthetic-NeRF dataset [25]. All scores for the baseline methods are directly taken from their respective published papers, whenever available. We show the training time, model size, PSNR, and rendering FPS (Frames per second).

| Method | Steps | Time | Size (MB) | PSNR ↑ | SSIM ↑ | LPIPS ↓ | FPS |
|---|---|---|---|---|---|---|---|
| NeRF [25] | 300k | 35.0h | 1.25 | 31.01 | 0.947 | 0.081 | - |
| Plenoxels [12] | 128k | 11.4m | 194.50 | 31.71 | 0.958 | 0.049 | - |
| PlenOctrees [39] | 200k | 15.0h | 1976.30 | 31.71 | 0.958 | - | - |
| DVGO [34] | 30k | 15.0m | 153.00 | 31.95 | 0.960 | 0.053 | - |
| Point-NeRF [37] | 200k | 5.5h | 27.74 | 33.31 | 0.962 | 0.049 | - |
| InstantNGP [26] | 30k | 3.9m | 11.64 | 32.59 | 0.960 | - | - |
| TensoRF-CP [9] | 30k | 25.2m | 0.98 | 31.56 | 0.949 | 0.076 | <30 |
| TensoRF-VM [9] | 30k | 17.4m | 71.80 | 33.14 | 0.963 | 0.047 | <30 |
| 3D GS [17] | 30k | 6.1m | 68.88 | 33.31 | 0.966 | - | 345.8 |
| Strivec [13] | 30k | 34.4m | 28.28 | 33.24 | 0.963 | 0.046 | <30 |
| Ours-CP-16 | 30k | 15.0m | 6.06 | 32.42 | 0.964 | 0.040 | 237.4 |
| Ours-VM-16 | 30k | 30.0m | 28.75 | 33.24 | 0.967 | 0.034 | 275.5 |

As illustrated in Fig. 3, utilizing factorized coordinates led to a significant reduction in the number of parameters required for representing 3D coordinates. Remarkably, it requires less than 10% of the original point cloud parameters to depict the contour of the object. However, this method introduced a certain level of blockiness in the approximation, which is inherent to the approach. To address this, we suggest the integration of learnable opacities (Secs. 3.3.1 and 3.3.2) and masks (Sec. 3.5), which could effectively mitigate this blockiness.

## 4.2 Experimental Settings

We implemented our F-3DGS model in PyTorch [29] framework with the original 3DGS [17] CUDA kernels for rasterization. In experiments, we used the default training hyperparameters of 3DGS [17] and added additional hyperparameters required for the F-3DGS model. To achieve the coarse-to-fine scene geometry reconstruction, we first jointly train factorized coordinates and representations. However, we found that joint training often leads to unstable optimization after a certain number of iterations. Therefore, after 20K iterations, we fixed the coordinates and only optimized the factorized attributes of Gaussians. For CP decomposition, we employed MLPs with a single hidden layer consisting of 128 hidden units. For VM decomposition, we utilized MLPs with two hidden layers. We used the Adam optimizer with initial learning rates of 0.02 for 3D Gaussians' tensor factors and 0.001 for the MLP decoder. As for scale coefficients, we did not use any activation function.

## 4.3 Comparison

In this subsection, we evaluated F-3DGS using two real-world datasets (objects from Tanks&Temples [18] and indoor scenes from Mip-NeRF 360 [3]) as well as one synthetic dataset (synthetic-NeRF [25]).

*Qualitative Result.* For a qualitative evaluation, we selected two distinct scenes from the synthetic-NeRF dataset. As Fig. 5 shows, we obtain similar visual quality on the Ship object only with 4-7MB. Similarly, for the Mic object, our VM-48 model requires 16MB storage, while the original 3DGS costs 40-50MB storage.

Table 2: Performance comparison on the Tanks&Temples dataset.

| | Size(MB) | PSNR ↑ | SSIM ↑ | LPIPS ↓ | FPS |
|---|---|---|---|---|---|
| NeRF[25] | - | 25.78 | 0.864 | 0.198 | - |
| NSVF[22] | - | 28.40 | 0.900 | 0.153 | - |
| TensoRF-CP[9] | 3.9 | 27.59 | 0.897 | 0.144 | <20 |
| TensoRF-VM[9] | 71.8 | 28.56 | 0.920 | 0.125 | <20 |
| Strivec-48[13] | 54.08 | 28.70 | 0.924 | 0.113 | <20 |
| 3DGS[17] | 105.15 | 30.88 | - | - | 170.8 |
| Ours-CP-16 | 10.94 | **30.29** | **0.957** | **0.061** | 138.8 |

Table 3: Performance comparison on Mip-360 indoor scenes.

| | room | kitchen | counter | bonsai | Model size (avg) |
|---|---|---|---|---|---|
| Strivec [13] | 28.11 | - | - | - | 12.6MB |
| DVGO [34] | 28.35 | - | - | - | 5.1GB |
| 3DGS [17] | 31.7 | 30.32 | 28.70 | 31.98 | 334.75MB |
| Ours | 30.84 | 30.14 | 28.14 | 31.23 | 70.50MB |

To analyze more deeply, we selected six objects from the synthetic-NeRF dataset to illustrate the distribution of Gaussians and rendered images using both our method and the original 3DGS (Fig. 4). Our F-3DGS only requires 10% of storage while performing accurate reconstruction. For example, on the Drum surface, our F-3DGS points distribute an aligned axis in each set of factorization coordinates, while the original 3DGS points are unordered. This comparison reveals that our factorized representation enables an aligned and denser Gaussian distribution. By employing the factorization of 3D Gaussians, our model captures and represents the flat surfaces of objects such as the Hotdog and Drums efficiently, thereby achieving compact model sizes without compromising on fidelity. Attributing to the learnable scale parameters of Gaussian points, our model can also obtain high fidelity on smooth surfaces, such as the Ship object. The density of F-3DGS can reconstruct more detailed objects without increasing the number of parameters, showcasing the

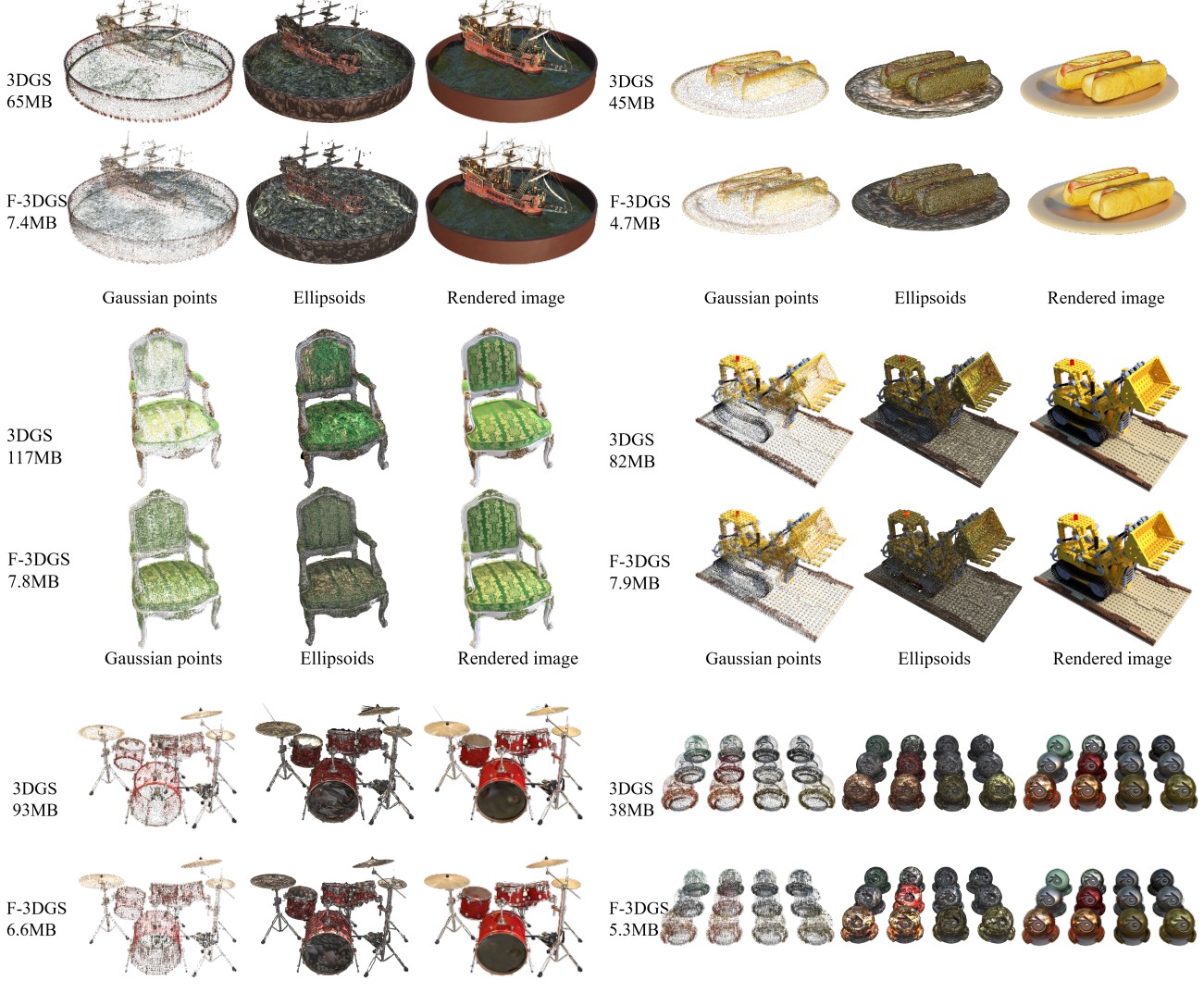

**Figure 4: Visualiztion of F-3DGS and 3DGS. These visualize Gaussian points, ellipsoids, and rendered images of six objects. We present the storage requirements for our CP-16 F-3DGS.**

efficacy of our method in achieving high-quality renderings with a reduced storage requirement.

*Quantitative Result.* Tab. 1 shows the quantitative results evaluated on the synthetic-NeRF dataset. Our approach consistently reduces the storage requirements while maintaining fast rendering speed. In particular, the approach obtained the equivalent standard or even surpassed TensoRF [9] and Strivec [13]. With 6.06 MB storage costs, our CP model can achieve 32.42 on PSNR for synthetic-NeRF dataset [25].

In addition, further experiments are conducted using Tanks&Temples and 360 indoor scenes datasets. Note that we achieve a 10× compression ratio and the same rendering speed on the Tanks&Temples dataset in Tab.2. And Tab.3 shows our quantitative result on 360 indoor scenes.

### 4.4 Ablation Study

*Initialization.* As addressed in Sec. 3.4, the initialization is an important factor in achieving high rendering quality in both 3D Gaussian Splatting (3DGS) and our proposed method. As shown in Tab. 4, using random initialization significantly deters the quality down to 26.06, which demonstrates the significance of our initialization method.

*Factorization of Non-coordinate Attributes.* The original 3DGS produces a significant amount of Gaussians, and moreover, some nearby

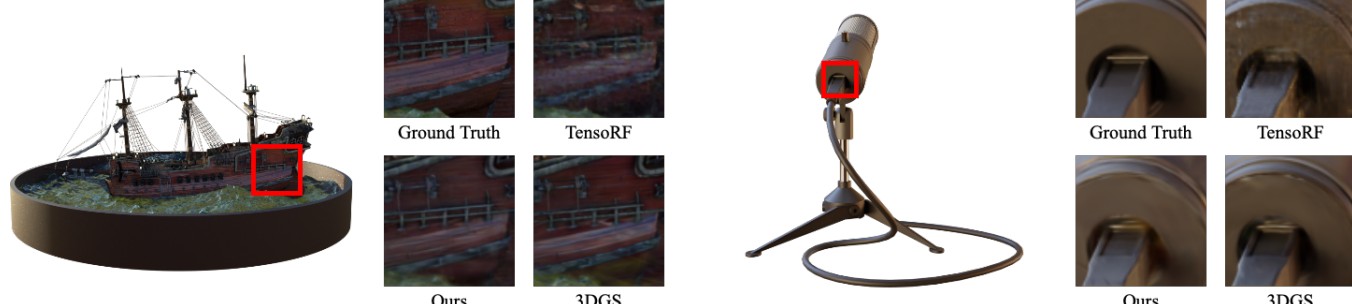

**Figure 5: Qualitative results. For our method, we used CP with a *d* of 16 in the case of our model, which is about 4–7 MB. For TensoRF, we visualized VM-48, which is about 16 MB. For 3DGS, we used the original 3DGS of 40–50 MB.**

**Table 4: Ablation study results using CP-based F-3DGS with a *d* of 48.**

|  | lego | chair | drums | ficus | hotdog | materials | average / storage / FPS |
|---|---|---|---|---|---|---|---|
| random initialization | 28.32 | 27.64 | 21.32 | 19.68 | 32.98 | 24.45 | 26.06 / 14MB / - |
| w/o color,opacity | 33.34 | 34.47 | 26.13 | 35.31 | 37.29 | 30.07 | 32.84 / 105MB / - |
| w/o scale,rotation | 35.05 | 29.66 | 25.90 | 34.89 | 36.94 | 29.66 | 32.85 / 30MB / - |
| w/o masking | 35.12 | 30.09 | 26.01 | 34.72 | 37.12 | 30.09 | 32.90 / 14MB / 125.6 |
| Full | 35.14 | 30.13 | 25.99 | 34.87 | 37.12 | 30.13 | 32.93 / 14MB / 237.4 |

Gaussians share similar attributes. To address this problem, the proposed factorization of color and opacity achieves a reduction in storage requirements by approximately 70%, and factorization of scale and rotation by 25% while maintaining the rendered quality. As Tab. 4 shows the storage could be compressed to 14MB with our F-3DGS.

*Learnable Masking.* It is crucial that densification scheme in 3DGS [17] is not feasible in our structured factorization coordinates approach. However, the redundant Gaussians in our factorized coordinates also need to be eliminated. Considering real-time rendering and training speed, we utilize the binary mask to prune out redundant and unessential 3D Gaussians to accelerate while only increasing a little storage to store mask parameters. The rendering speed is nearly 90% increase than the method without masking in Tab. 4.

**Table 5: Comparison of the model sizes (in MBs) and average PSNRs of CP and VM decomposition on synthetic-NeRF dataset [25] with different numbers of components, optimized for 40k steps.**

|  | *d* | Size (MB) | PSNR |
|---|---|---|---|
| Factorize-CP | 16 | 6.06 | 32.42 |
|  | 24 | 8.10 | 32.56 |
|  | 48 | 13.97 | 32.88 |
|  | 96 | 25.83 | 33.13 |
| Factorize-VM | 8 | 16.20 | 32.80 |
|  | 16 | 28.75 | 33.14 |
|  | 24 | 42.13 | 33.21 |
|  | 48 | 81.46 | 33.33 |

*The Size of Learnable Feature Vectors.* As mentioned in Secs. 3.3.1 and 3.3.2, our method introduces a hyperparameter *d* that controls the number of learnable feature dimensions. We evaluate our Factorize-3DGS on the synthetic-NeRF dataset [25] using both CP and VM decompositions with different numbers of *d*.

As shown in Tab. 5, the larger *d* enhances the 3D reconstruction performance, simultaneously increasing the overall storage requirements. Both Factorize-CP and Factorize-VM achieve consistently better rendering quality with more feature dimensions. Factorize-CP achieves a more compact model with high performance by using our factorized method. Ours-CP-16 achieves 32.43 PSNR with 6.06 MB, outperforming TensoRF-CP (see Tab. 1). Also, F-3DGS using VM-16 achieves the visual quality of TensoRF-VM (33.14 PSNR) but requires significantly less storage only 28.75 MB, compared to 71.80 MB needed by TensoRF-VM.

## 5 CONCLUSION

In this paper, we have proposed a novel Factorized 3D Gaussian Splatting (F-3DGS), effectively addressing the computational and resource constraints in neural rendering applications. Our method integrates tensor factorization techniques, significantly reducing the storage requirements while maintaining image quality compared to 3DGS, as evidenced by our extensive experiments across various datasets. We believe this newly introduced representation opens a new direction for further research. Compared to traditional matrix and tensor factorization, the proposed approach does not involve any pre-defined grid representations. This will enable us to model very large and unbounded scenes more efficiently since we can ignore large empty parts in the scene, as opposed to grid-based approaches that require additional sparsification techniques.

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
