# OpenReview forum: "F-3DGS: Factorized Coordinates and Representations for 3D Gaussian Splatting"
_acmmm.org/ACMMM/2024/Conference — MM2024 Poster_

### Official Review · Reviewer_ngLn · 2024-04-28

**Rating:** 4
**Confidence:** 3

**Summary:**

This paper proposes a compact representation for 3D Gaussians, namely F-3DGS. In the proposed method, 3D Gaussians are compactly represented by factorized coordinates and latent embeddings associated with the coordinates. Then, 3D Gaussian attributes are generated from latent embeddings of given factorized coordinates. Meanwhile, learnable masks are applied to coordinate sets to remove redundant 3D Gaussians.

**Strengths:**

1. The proposed factorized coordinate representation of 3D Gaussians is novel and interesting. This is because factorized representation has been proven effective in NeRF slimming, while its application in 3D Gaussian splatting has not been fully studied. This paper successfully uses spatial decomposition to compactly represent 3D Gaussian primitives as factorized grids and effectively recover primitive parameters from grid features.
2. The motivation of the proposed method is reasonable. This is because there are redundant primitives in the model of vanilla 3DGS, which can be removed by the proposed masking mechanism. In addition, there are correlations between spatially adjacent Gaussian primitives, which can be implicitly captured by the factorized grid to improve compactness.
3. The performance of the proposed method well demonstrates its effectiveness. That is, the proposed method is able to achieve 10x size reduction compared to vanilla 3DGS.

**Limitations:**

1. Hyper-parameters in Sec 3.4 should be specified for each dataset. For example, the hyper-parameter lambda should be specified for each dataset, and the average interval of bins are also should be specified.
2. I would like to know why you did not report the performance of VM decomposition on the Tanks & Temples dataset. Besides, which factorization paradigm do the results on Mip-360 correspond to?
3. I have some concerns about the experimental results. For example, the size of 3DGS on the Tanks&Temples dataset exceeds 400MB, as reported in existing work, while the size reported in this paper is only 105.15MB. Therefore, the authors should explain the model size calculation method.
4. I would like to know why the model size does not decrease after adding the mask, as shown in Table 4. I would like to see more analysis of this phenomenon.

**Suitability:**

2

---

### Official Review · Reviewer_bsdo · 2024-05-24

**Rating:** 4
**Confidence:** 3

**Summary:**

This method proposes a 3D scene representation that factorizes both coordinates and 3D Gaussian Splatting (3DGS) features. By introducing structured coordinate and feature factorization, it significantly reduces storage requirements. Additionally, the method proposes an initialization approach that generates point clouds from a pre-trained 3DGS model, factorizes these point clouds, and then initializes the 3DGS. The method also introduces an adaptive mask to remove redundant Gaussians in empty spaces.

**Strengths:**

Experimental results demonstrate the effectiveness and superiority of this method by using structural similarity index (SSIM) and learned perceptual image patch similarity (LPIPS) metrics. This method makes it easier for 3DGS to fit structured objects and reduces a certain model size

**Limitations:**

Although this method can compress the model and improve structural attributes, its indicator improvement is not very competitive, and the training time is much longer than that of 3DGS.

**Suitability:**

3

---

### Official Review · Reviewer_kZdK · 2024-05-24

**Rating:** 3
**Confidence:** 4

**Summary:**

This paper proposes a novel framework based on factorized representation to reduce the storage requirements of 3D Gaussian splatting. Specifically, 3D coordinates are decomposed as the Cartesian product of three axes or unions of the Cartesian product of a plane and an axis. Then, scale, rotation, and learnable features are factorized by the outer product of each axis and utilized to further generate attributes of 3D Gaussian primitives. Experimental results demonstrate that the proposed method achieves around 10 times size reduction with comparable quality when compared to the original 3DGS.

**Strengths:**

1.	The introduction is well-written and easy to follow.
2.	The illustration of the factorized representation is interesting and technically sound.
3.	The experimental results show that the proposed method can significantly reduce the size of 3D Gaussian primitives without sacrificing rendering quality.

**Limitations:**

1.	In the experiment section, authors should compare other mentioned methods that lightweight 3DGS.
2.	What is the difference between TensoRF [9] and the proposed method, in addition to applying CP and VM factorization [9] to 3DGS? It would be better to explicitly state the special design in terms of 3DGS to highlight the novelty.
3.	Table 2 shows that 3DGS has 30.88 dB with 105.15MB. However, 3DGS presents 23 dB with 411MB on the same dataset in their paper. Please explain this discrepancy.
4.	What is the data type of factorized coordinates, integers or floating-point numbers? Could the masking strategy remove pseudo coordinates generated by the Cartesian product? Please explain why the size without masking is the same as the full proposed method in Table 4 in the ablation study.
5.	The authors generated point clouds from a pre-trained 3DGS model as initialization. It would be better to compare initialization from the original SfM points and the pre-trained points in the ablation study.
6.	It would be better to improve the illustration of the initialization part.

**Suitability:**

3

---

### Official Review · Reviewer_jU9E · 2024-05-25

**Rating:** 4
**Confidence:** 3

**Summary:**

This paper addresses the computational costs associated with 3D Gaussian Splatting (3DGS), which offers fast rendering but requires substantial storage. To mitigate this, the authors propose Factorized 3D Gaussian Splatting (F-3DGS), which reduces storage requirements while preserving image quality. F-3DGS approximates dense clusters of Gaussians with fewer Gaussians through efficient factorization techniques. This approach represents dense 3D Gaussians by encoding them with limited information along each axis, reducing storage costs while maintaining comparable quality in rendered images.

**Strengths:**

- This paper is written well and very easy to follow.
- The proposed idea is inspired by TensoRF and Strivec, the authors apply their idea to decompose the parameters of 3DGS, including color, scale, and rotation. This approach effectively reduces storage requirements while having a minimal impact on PNSR performance.
- The validation is comprehensive, and the source codes are provided by the authors.

**Limitations:**

- 3DGS is an explicit representation; unlike TensoRF, the memory footprint in 3DGS may not be able to be reduced. I would like to see more results about VRAM memory footprint during the training, inference, and rendering of the proposed method instead of only storage memory.
- The qualitative results make me think of recent works in 3DGS-based inverse rendering and surface reconstruction, for example, GaussianShader, GIR, 2D Gaussian Splatting, etc., by aligning the axis of 3D/2D Gaussian towards the screen and improving the efficiency of 3D/2D Gaussians. The authors are encouraged to discuss those works in the literature review.
- I would like to understand why the authors choose not to decompose positions (i.e., means) of 3D Gaussians, the authors could discuss it a bit.

**Suitability:**

2

---

### Meta-Review · Area_Chair_UPbZ · 2024-07-02

**Recommendation:** Accept (Poster)
**Confidence:** 4

**Metareview:**

This paper was reviewed by four experts in the field. The recommendations are Weak Accept, Borderline Accept, Borderline Accept, Weak Reject. The authors have addressed majorities of the concerns from reviewers about the experimental comparison. Based on this, the decision is to recommend the paper for acceptance to ACM Multimedia 2024.

Still, the reviewers did raise some valuable concerns, including 1) comparison to other lightweight models, and 2) discussing the trade-off of increasing training time and smaller model size. We recommend that the authors carefully read all reviewers' final feedback, and revise the manuscript as needed. We congratulate the authors on the acceptance of their paper!